# The Hydrophobic Extract of *Sorghum bicolor* (L. Moench) Enriched in Apigenin-Protected Rats against Aflatoxin B_1_-Associated Hepatorenal Derangement

**DOI:** 10.3390/molecules28073013

**Published:** 2023-03-28

**Authors:** Solomon E. Owumi, Blessing Ajakaiye, Adenike O. Akinwunmi, Sarah O. Nwozo, Adegboyega K. Oyelere

**Affiliations:** 1Cancer Research and Molecular Biology Laboratories, Department of Biochemistry, Faculty of Basic Medical Sciences, University of Ibadan, Ibadan 200005, Nigeria; 2Nutrition and Industrial Biochemistry Laboratories, Department of Biochemistry, Faculty of Basic Medical Sciences, University of Ibadan, Ibadan 200005, Nigeria; 3Department of Chemistry, Ekiti State University, Ado-Ekiti, Ekiti 360001, Nigeria; 4School of Chemistry & Biochemistry, Parker H. Petit Institute for Bioengineering and Bioscience, Georgia Institute of Technology, Atlanta, GA 30332, USA

**Keywords:** aflatoxin B_1_, hepatorenal toxicity, *Sorghum bicolor* sheath hydrophobic fraction, LC-MS and HPLC protocols, apigenin, redox and inflammatory balance

## Abstract

Aflatoxin B1 (AFB_1_) is a recalcitrant metabolite produced by fungi species, and due to its intoxications in animals and humans, it has been classified as a Group 1 carcinogen in humans. Preserving food products with *Sorghum bicolor* sheath can minimise the contamination of agricultural products and avert ill health occasioned by exposure to AFB_1_. The current study investigated the ameliorating effect of *Sorghum bicolor* sheath hydrophobic extract (SBE-HP) enriched in Apigenin (API) on the hepatorenal tissues of rats exposed to AFB1. The SBE-HP was characterised using TLC and LC-MS and was found to be enriched in Apigenin and its methylated analogues. The study used adult male rats divided into four experimental cohorts co-treated with AFB1 (50 µg/kg) and SBE-HP (5 and 10 mg/kg) for 28 days. Biochemical, enzyme-linked immunosorbent assays (ELISA) and histological staining were used to examine biomarkers of hepatorenal function, oxidative status, inflammation and apoptosis, and hepatorenal tissue histo-architectural alterations. Data were analysed using GraphPad Prism 8.3.0, an independent *t*-test, and a one-way analysis of variance. Co-treatment with SBE-HP ameliorated an upsurge in the biomarkers of hepatorenal functionality in the sera of rats, reduced the alterations in redox balance, resolved inflammation, inhibited apoptosis, and preserved the histological features of the liver and kidney of rats exposed to AFB_1_. SBE-HP-containing API is an excellent antioxidant regiment. It can amply prevent the induction of oxidative stress, inflammation, and apoptosis in the hepatorenal system of rats. Therefore, supplementing animal feeds and human foods with SBE-HP enriched in Apigenin may reduce the burden of AFB1 intoxication in developing countries with a shortage of effective antifungal agents.

## 1. Introduction

Aflatoxins (AFs) are mycotoxins produced by fungi species, especially the genus *Aspergillus*, including *A. flavus*, *A. parasiticus*, *A. nomius*, *A. niger*, and *A. pseudotamarii* [1,2,3]. The biosynthesis of AFs by fungi is driven under certain climatic conditions, including a temperature range of 25–35 °C and relative humidity of 80 to 100%, as typical in Sub-Saharan Africa [2,4], resulting in food spoilage [2,5,6] and significant economic loss [7]. AFs are classified, based on absorbance wavelengths and chromatographic mobility, into aflatoxin B_1_ (AFB_1_), aflatoxin B_2_ (AFB_2_), aflatoxin G_1_ (AFG_1_), and aflatoxin G_2_ (AFG_2_). In this classification guideline, the letters ‘B’ and ‘G’ are delineated as blue and green florescent colours emitted under UV light on thin-layer chromatographic plates. In contrast, subscripts 1 and 2 refer to the major and minor compounds, respectively [8]. Based on toxicity, the potency of AFs to mediate mutagenicity, carcinogenicity, and teratogenicity in humans and animals is in the order: AFB_1_ > AFG_1_ > AFB_1_ > AFG_2_ [9,10,11]. The class AFB_1_ is the most toxic of all the known aflatoxins, and it has been recognised by the World Health Organization (WHO) and the International Agency for Research on Cancer (IARC) as a group I carcinogenic hazard to humans [12]. The LD_50_ (lethal dose) of AFB_1_ highly depends on the species of organisms, with organisms such as rats, dogs, sheep, and monkeys being susceptible, while chickens and mice are resistant species [13,14]. Initially, it was claimed that humans could tolerate exposure to AFB_1_. However, recent carcinogenic potency established by the Joint FOA/WHO Expert Committee on Food Additives (JECFA) in Africa defined a maximum tolerable limit. It revealed that exposure to AFB_1_ is attributed to growth retardation, malnutrition, and immune suppression [15,16].

AFB_1_ toxicity depends on the absorption routes, distribution pattern, differences in the expression of CPY450 isoforms, and excretory pathways. AFB_1_ is absorbed and distributed to the enterocytes and hepatocytes, where different CYP_450_ enzymes act upon AFB_1_, including CYP3A4, 1A2, 2E6, and 3A5 [17,18] and NADPH-dependent reductase [19]. Specifically, NADPH-dependent reductase converts AFB_1_ into aflatoxicol (AFL), CYP3A4 and CYP1A2 convert AFB_1_ into aflatoxin Q_1_ (AFQ_1_), CYP1A1 and CYP1A2 convert AFB_1_ into aflatoxin M_1_ (AFM_1_), CYP3A4 converts AFB_1_ to aflatoxin P_1_ (AFP_1_) through an O-demethylation reaction, and CYP1A1, CYP1A2, CYP2E6, CPY3A4, and CYP3A5 convert AFB_1_ to aflatoxin B1-8, 9-epoxide (AFBO), an extremely reactive and toxic metabolite (Figure 1). Unlike AFBO, AFL, AFQ_1_, AFM, and AFP_1_ are clinically insignificant as they are not heavily implicated in mutagenicity, carcinogenicity, and teratogenicity. The mechanisms of AFBO toxicity are known to be through the formation of protein adducts, DNA adducts, and lipid peroxidation. Uncontrolled degradation of functional biological molecules following unfettered exposure to AFB_1_ has been observed to deplete the redox buffering system of rats, thereby predisposing cells to oxidative stress, inflammation, and programmed cell death [20,21,22]. Nonetheless, there is a safe pathway for removing AFBO from the enterocytes and hepatocytes via the second phase of biotransformation mediated by glutathione S-transferase (GST). GST is known to mediate the conjugation of AFBO with GSH to form the AFB1-*S*-G complex. This complex may bind to the 190-kDa multi-drug resistant protein (MRP) and the 170-kDa P-glycoprotein [23] and is extruded from the hepatocytes and enterocytes into bile and urine. Depleting GST and other innate antioxidant defence systems activities is known to increase the concentration of AFBO in the cells, leading to enhanced oxidative stress, oxidative DNA damage, chronic inflammation, and apoptosis in the hepatocytes and nephrons of rats [22,24].

Previous studies reported the harmful effects of AFB_1_ on the reproductive system [21], the endocrine system [25], the central nervous system [26], the cardiovascular system [27], and the immune system [28]. These reports reiterate that direct exposure to AFB_1_ could promote the development of disease and infection in animals and humans, hence the need to mitigate indiscriminate exposure to AFB_1_ and its associated health sequela.

Sorghum (*Sorghum bicolor* (L.) Moench is majorly cultivated as food in the semi-arid tropical regions of Asia and Africa [29]. Sorghum is a rich source of bioactive phytochemicals, including the two major 3-deoxyanthocyanidins, apigeninidin [30] and luteolinidin [31], and flavonoids such as apigenin, which are known to promote good health. Compounds extracted from Sorghum have been shown in in vivo and in vitro studies to have therapeutic effects against non-infectious disorders such as obesity, diabetes, dyslipidemia, cardiovascular disease, cancer, and hypertension [30,32]. Extracts from Sorghum enriched in certain phytochemicals such as apigenin (API), luteolin, luteolinidin, and apigeninidin in *S. bicolor* extracts have been shown to reduce oxidative stress by upregulating antioxidant enzymes while reducing the generation of reactive oxygen species (ROS) [30,33,34,35]. Based on the structural–activity relationship (SAR), the abilities of these compounds to stall oxidative stress may be attributed to their possession of several aromatic rings and phenolic groups [36]. These properties endow the leaf sheath extract of *S. bicolor* with high antioxidant potential and increase its free radical scavenging activity against ROS such as hydroxyl radicals and superoxide anions [37], one of the chief culprits of AFB_1_-induced hepatorenal toxicities. Experimental studies revealed that extracts of *Sorghum bicolor* (*S. bicolor*) significantly diminished ROS production by immune cells [37]. This report demonstrates the anti-inflammatory and immunomodulatory properties. Recently our laboratory developed a method to separate phytochemicals in *S. bicolor* into apigeninidin-enriched alcoholic extracts and a hydrophobic (CH_2_Cl_2_) extract (*S. bicolor* extract—SBE-HP). We observed that apigeninidin-enriched extracts of the plant protected the liver and kidney of male rats from AFB_1_-mediated increase in hepatorenal dysfunctional molecules, generation of ROS, pro-inflammatory molecules, and apoptotic proteins [30]. To further probe the ethnomedicinal benefits of the SBE-HP, we herein characterised the components of this hydrophobic extract using chromatographic and LC-MS techniques. We found SBE-HP devoid of apigeninidin and enriched in apigenin and apigenin analogues. Subsequently, we investigated the potential impact of this apigenin-rich SBE-HP in abating liver and kidney derangements following sub-acute exposure to AFB_1_ in male Wistar rats. The intention is to determine if SBE-HP could abrogate the AFB_1_-induced liver and kidney toxicities via antioxidative, anti-inflammatory, and apoptotic mechanisms. To this end, we treated rats with AFB_1_ and SBE-HP for 28 days and evaluated the effect of this treatment on endogenous antioxidant defence systems and levels of anti-inflammatory cytokines. We observed that SBE-HP improved endogenous antioxidant defence systems and upregulated anti-inflammatory cytokines while reversing the basal concentrations of hepatorenal dysfunction molecules, ROS, pro-inflammatory and apoptotic mediators, and histological lesions in the liver and kidney of rats challenged with AFB_1_.

## 2. Results

### 2.1. Characterisation of the Key Components in SBE-HP

The identities of the key components in SBE-HP, a CH_2_Cl_2_ extract of *S. bicolor*, were probed using a combination of TLC and LC-MS analyses. TLC using CH_2_Cl_2_/MeOH 12:1 revealed that the components of SBE_HP clustered into three spots with retention factors (*Rf*) of 0.27, 0.83, and approx. 1 (migrated with the solvent front). The *Rf* 0.27 is identical to API in this solvent system (Figure 2A), suggesting that the SBE-HP components with this Rf contain API and close analogues. Subsequent LC-MS analysis revealed that the main components of SBE-HP are API, apigenin O-methyl ethers—acacetin and glycitein (or genkwanin), apigenin dimethyl ethers, hispidulin or luteolin methyl ethers, luteolin dimethyl ether, naringenin, 7-O-methyl apigeninidin, and protocatechuic acid methyl ester (Figure 2B).

### 2.2. SBE-HP Prevented AFB1-Orchestrated Alterations in Body Weight and Organosomatic Indices, Rat Survivability, and Biomarkers of Hepatorenal Functions in Rats

The preventive effects of SBE-HP on AFB_1_-induced hepatorenal derangements in adult male Wistar rats were investigated in the current study. We explored the abrogative potential of SBE-HP on body weight and organosomatic indices (Figure 3B), overall survival (Figure 3C), and biomarkers of the functions and integrity of the hepatorenal system (Figure 4) in rats. Our results showed that the mean final body weights of the experimental animals increased significantly (*p* < 0.05) in all groups compared to the initial body weight of rats. However, there was no significant difference (*p >* 0.05) in the mean organ weight and relative organ weight of rats. Compared to the control, AFB_1_ slightly decreased the mean weight change of rats. However, this was abated in the group treated with only 5 mg/kg SBE-HP (Figure 3B). Despite the slight decrease in organ-to-somatic ratio and mean body weights, rats had no mortality during the study period, indicating 100% overall survival (Figure 3C). In addition, the administration of AFB_1_ significantly increased (*p* < 0.0001) the serum activities and levels of ALT, AST, ALP, urea, and creatinine compared to the control, indicative of hepatorenal toxicities. However, this was counterbalanced by SBE-HP at 5 and 10 mg/kg. At 5 mg/kg, SBE-HP significantly decreased the serum activities and levels of ALT (*p* < 0.001), AST (*p* = 0.0387), ALP (*p* < 0.0001), urea (*p* = 0.0124), and creatinine (*p* < 0.0001) compared to the cohort of rats treated with alone AFB_1_. The restoration in the serum concentrations of these hepatic and renal function biomarkers was significantly prominent (*p* < 0.0001) in a cohort of rats co-treated with 10 mg/kg SBE-HP compared to the AFB_1_ alone group (Figure 4).

### 2.3. SBE-HP Mitigates Oxidative Stress and Restores Antioxidant Activities and Levels in the Liver and Kidney of AFB_1_-Treated Rats

The induction of oxidative stress is a probable mechanism of action of AFB_1_-induced hepatic and renal derangements. To this end, we probed the mitigating effect of SBE-HP against AFB_1_-mediated oxidative stress in rats’ hepatic and renal tissues, as presented in Figure 5, Figure 6 and Figure 7. In comparison to the control, our results showed that AFB_1_ significantly decreased (*p* < 0.0001) the hepatic activities of SOD, CAT, and GPx as well as the renal activities of SOD (*p* = 0.0084), CAT (*p* < 0.0001), and GPx (*p* < 0.0001) in rats, denoting the induction of oxidative stress. However, the administration of SBE-HP at 5 and 10 m/kg restores the activities of these antioxidants. At 5 mg/kg, SBE-HP increased the activities of SOD (*p* < 0.0001), CAT (*p* > 0.6186), and GPx (*p >* 0.9999) compared to the AFB_1_ alone group. However, at a higher concentration of SBE-HP (10 mg/kg), the activities of SOD, CAT, and GPx were prominent (*p* < 0.0001) relative to AFB_1_ alone treated rats (Figure 5). Furthermore, our results divulged that AFB_1_ markedly waned (*p* < 0.0001) the activities of GST (a phase-2 antioxidant molecule) and the levels of GSH and TSH in the liver and kidney of rats compared to the untreated group. In contrast, the co-ingestion of SBE-HP at the tested doses resulted in a positive trend in GST activity and GSH and TSH levels in the liver and kidney of rats. Rats co-treated with 5 mg/kg SBE-HP showed increased hepatic GST activity (*p* = 0.0501) and GSH (*p* < 0.0001) and TSH (*p* < 0.0001) levels. Renal GST activity (*p* = 0.0008) and GSH (*p >* 0.9999) and TSH (*p = 0.0007*) levels were also increased compared to the AFB_1_ alone treated group. The observed effects on GST activity and GSH and TSH levels were further reinforced (*p* < 0.001) in cohorts of rats co-treated with 10 mg/kg in the liver and kidney tissues of rats relative to the AFB_1_ alone treated rats (Figure 6). In addition, for oxidative stress to manifest in rats exposed to toxicants, there must be an imbalance between antioxidants and pro-oxidants in favour of pro-oxidants. In the current study, AFB_1_ significantly elevated (*p* < 0.0001) the hepatic and renal levels of LPO and RONS compared to the control. Nevertheless, co-treatment with SBE-HP offset this effect, thus scavenging available free radicals in the liver and kidney of rats. Specifically, co-treatment at 5 mg/kg waned the hepatic levels of LPO (*p* = 0.0070) and RONS (*p* < 0.0001) as well as significantly decreased (*p* < 0.0001) the renal levels of LPO and RONS relative to the AFB_1_ treated group. Furthermore, at 10 mg/kg, the decline in the levels of LPO and RONS was significantly prominent (*p* < 0.0001) compared to the AFB_1_ alone group in the liver and the kidney tissues, indicating that the extract mediates the antioxidative effect (Figure 7).

### 2.4. SBE-HP Resolves Inflammation and Abrogates Apoptosis in the Liver and Kidney of AFB_1_-Treated Rats

In the current study, we hypothesise that unquenched free radicals may trigger the release of pro-inflammatory molecules and initiate the programmed cell death pathways through extrinsic or intrinsic pathways. We, therefore, probe the ameliorative effect of SBE-HP against AFB_1_-mediated inflammation and apoptosis, as shown in Figure 8, Figure 9 and Figure 10. The treatment of rats with AFB_1_ significantly increased pro-inflammatory mediators, including XO, MPO, and NO, compared to the untreated group (*p* < 0.0001). However, co-treatment with SBE-HP counteracted the upsurge in pro-inflammatory mediators at the test doses. Specifically, 5 and 10 mg/kg of SBE-HP markedly waned (*p* < 0.0001) the activities of XO and MPO and the level of NO in the liver and kidney compared to the cohort of rats treated with AFB_1_, with the higher dose (10 mg/kg) exhibiting higher anti-inflammatory property (Figure 8). We hypothesise that the pro-inflammatory and anti-inflammatory cytokines ratio may induce hepatorenal toxicity. To this end, we probe the effect of SBE-HP in lowering the proportion of pro-inflammatory cytokines to anti-inflammatory in rats exposed to AFB_1_. Compared to the control, the rat cohort treated with AFB_1_ alone exhibited increased levels of IL-1β in the liver (*p* = 0.0031) and in the kidney (*p* < 0.0001) while reducing the level of IL-10 in the liver (*p* = 0.0014) and the kidney (*p* = 0.0003). In contrast, cohorts of rats co-treated with SBE-HP at the test doses decreased the hepatic and renal levels of IL-1β while elevating IL-10. At 5 mg/kg, SBE-HP reduced the level of IL-1β in the liver (*p* = 0.2208) and kidney (*p* = 0.0018) while increasing the level of IL-10 in the liver (*p* = 0.024) and kidney (*p* = 0.0043) of rats compared to AFB_1_ alone treated rats. The anti-inflammatory effect of SBE-HP was more prominent in cohorts of animals co-treated at 10 mg/kg, as this dose decreased the level of IL-1β in the liver (*p* = 0049) and kidney (*p* < 0.0001) while increasing the level of IL-10 in the liver (*p* < 0.0001) and kidney (0.0004). SBE-HP at both doses lowered IL-1β/IL-10, favouring anti-inflammatory cytokines (Figure 9). In addition, we hypothesised that if oxidative stress and inflammation are not resolved, they may drive apoptosis and therefore probe that SBE-HP can inhibit the ability of AFB_1_ to orchestrate apoptosis in the liver and kidney of rats. Our results showed that AFB_1_, compared to the untreated group, triggered apoptosis by increasing the hepatic activities of caspase-9 and caspase-3 (*p* < 0.0001) as well as renal activities of caspase-9 (*p* = 0.0018) and caspase-3 (*p* < 0.0001) compared to the control. In contrast, SBE-HP at the tested doses prevented apoptosis in the liver and kidney of rats. Cohorts of rats co-treated with 5 mg/kg SBE-HP decreased hepatic activities of caspase-9 (*p* = 0.0062) and caspase-3 (*p* = 0.0008) and renal activities of caspase-9 (*p* = 0.0283) and caspase-3 (*p* = 0.0020) compared to the AFB_1_ alone treated rats. This effect was enhanced in cohorts of rats co-treated with 10 mg/kg SBE-HP as this decreased the hepatic activities of caspase-9 and caspase-3 (*p* < 0.0001) as well as hepatic activities of caspase-9 (*p* = 0.0012) and caspase-3 (*p* < 0.0001) relative to the AFB_1_ alone treated rats (Figure 10).

### 2.5. SBE-HP Abrogates Histological Lesions in the Liver and Kidney of AFB_1_-Exposed Rats

Finally, we probe that exposure to AFB_1_ after 28 days can orchestrate alterations in the architectural structures of the kidney and liver tissues and that co-treatment with SBE-HP can reduce the burden of AFB_1_-hepatorenal toxicity and preserve the architectural forms of the rat kidney and liver, as presented in Figure 11a,b. In comparison to the untreated group, cohorts of rats treated with AFB_1_ manifested dispersed glomerular messangialisation and infiltration of intraglomerular mesangial cells into the cortex in the kidney tissues (Figure 11a), as well as focal congestion, trafficking, and infiltration of Kupffer cells into zone 2, slight ballooning and degeneration of the liver cells and micro-vesicular steatosis in the liver tissues (Figure 11b). However, SBE-HP at 5 and 10 mg/kg minimised AFB1-mediated toxicities and preserved the histoarchitectural structures of the liver and kidney tissues, approximating histo-architecture like the control.

## 3. Discussion

Exposure of animals and humans to AFB_1_ continues to be of severe concern in semi-arid tropical regions of Asia and Africa. Accidental ingestion of AFB_1_ through contaminated food products can cause toxicity to organs such as the liver, kidney, hypothalamus, testis, epididymis, and heart [21,22,24,27]. The concomitant effects of AFB_1_-induced toxicities include growth retardation, malnutrition, and immune suppression [15,16]. Efforts to reduce AFB_1_ contamination in food products have continually waned due to the recalcitrant nature of the fungal species that mediate the biosynthesis of mycotoxins [38,39] and the induction of metabolic pathways for the breakdown of AFB_1_ [30,33,40]. Bioactive compounds in plants can remediate AFB_1_-mediated toxicities by inhibiting ROS production, resolution of inflammation, and preventing programmed cell death [41,42]. To further narrow the gap in the search for potential drug regimens that can effectively lessen the side events associated with AFB_1_ toxicity, the current study explores the ameliorative effects of the SBE-HP against AFB_1_-mediated hepatorenal derangements. The results show slight alterations in mean body weight and organosomatic indices and a significant increase (*p* < 0.05) in the activities of liver and kidney function biomarkers (ALT, AST, ALP, urea, and creatinine). There was a considerable decrease (*p* < 0.05) in the levels of antioxidant (SOD, CAT, GPx, GST, GSH, and TSH) and anti-inflammatory (IL-10) biomarkers; a significant increase (*p* < 0.05) in the levels of oxidative stress (MDA, RONS), pro-inflammatory (XO, MPO, NO, and IL-1β), and apoptotic (caspase-9 and -3) biomarkers; atypical histological structures observed in AFB_1_ alone treated group were reversed, counteracted, resolved, and refurbished by co-treatment with 5 and 10 mg/kg SBE-HP. This study established three definitive mechanisms for AFB_1_ toxicities: high ROS generation, dampening the innate antioxidant defences, and activating pro-inflammatory molecules to activate, expand, and differentiate pro-inflammatory cells. Collectively, these caused the release of pro-inflammatory cells and increased the ratio of pro-inflammation to anti-inflammation in the hepatorenal system, resulting in the induction of apoptosis via the intrinsic or extrinsic pathway.

Oxidative stress is established when an imbalance between the antioxidant defence system and the pro-oxidants occurs [43,44]. AFB_1_ is a known hepatoxic and nephrotoxic agent in humans and experimental animals [22,40,45]. AFB_1_ toxicity stems from the activities of CYP540 isoforms, which mediate the bioactivation of AFB_1_ into a toxic metabolite, AFBO [17,18]. Continual bioactivation of AFB_1_ into AFBO by these enzymes releases superoxide anion radicals (O2^.-^) into the hepatorenal system, thereby orchestrating SOD, an enzyme known to detoxify (O2^.-^) into hydrogen peroxide (H_2_O_2_), a less toxic free radical. The accumulation of H_2_O_2_ rouses the catalase activity to degrade the hepatic and renal levels of H_2_O_2_ into molecular water. GPx can detoxify H_2_O_2_ in the presence of GSH or TSH. These innate antioxidant defence systems, including SOD, CAT, GPx, GSH, and TSH, were diminished, as seen in the current study. Concomitantly, free radicals accumulate in the hepatocytes and nephrons and trigger the formation of hydroxyl radicals (HO^-^) in the presence of ferrous ions (Fe^2+^) in either the Haber–Weiss or Fenton reactions. These radicals interact with important biological molecules such as membrane lipids, proteins, and nucleic acids, triggering lipid peroxidation, as observed in this study. We observed that AFB1 significantly increased the hepatic and renal levels of MDA, a marker of lipid peroxidation and protein cross-link and DNA and RNA damage established in other studies [22,40]. These biochemical reactions evolve more ROS and RNS in the liver and kidney cells, further hampering the capacity of the innate antioxidant defence systems to mop up hepatic and renal free radicals.

As observed in this study following treatment with AFB1, a significant increase in RONS levels in tissues is correlated to hepatorenal toxicities [46]. Nevertheless, the toxicities associated with AFB_1_ and AFBO can be averted by the phase-2 antioxidant defence system via the expression of GST. GST interaction with AFBO mediated by GSH and other sundry enzymes forms the AFB1-*S*-G complex. This complex binds to MRP or p-glycoprotein and is excreted in the bile and urine. This study shows that suppressing GST’s hepatic and renal activities further exposes the target organs to ROS-mediated toxicities, as more AFBOs are formed with the simultaneous evolution of abundant free radicals. Findings from previous studies on the oxidative/nitrosative stress-promoting effects of AFB_1_ in rats agree with our current observations [22,47,48].

Interestingly, we observed that cohorts of rats co-treated with SBE-HP at 5 and 10 mg/kg remediated oxidative/nitrosative stress in the liver and kidney tissues by significantly increasing the hepatic and renal levels of SOD, CAT, GPx, GSH, TSH, and GST. At the same time, SBE-HP markedly decreased LPO and RONS levels in rats’ livers and kidneys. The observed antioxidative effect is attributed to the unique antioxidant activities of the key compounds in SBE-HP. API, apigenin methyl ether analogues (acacetin, glycitein (or genkwanin), apigenin dimethyl ethers, hispidulin or luteolin methyl ethers, luteolin dimethyl ether, naringenin, 7-O-methyl apigeninidin, and protocatechuic acid methyl ester identified in SBE-HP are antioxidants that have been previously detected in *S. bicolor* [49]. We show that API inhibited the activity of CYP1A2, one of the CYP isoforms involved in the bioactivation of AFB_1_ into a toxic intermediate.

Unquenched oxidative/nitrosative stress could activate signalling pathways, especially NF-κB, a known regulator of inflammation [49], and other pro-inflammatory mediators, including MPO, XO, and NO [22]. MPO, a member of the heme peroxidase superfamily, is expressed in pro-inflammatory cells such as neutrophils and monocytes [50]. It mediates the biochemical reaction in which H_2_O_2_, in the presence of chlorine, is converted to hypochlorous acid (HOCl), which decomposes to release harmful free radicals, including ^1^O_2_ and OH. An increase in MPO activity during inflammation has been shown to drive oxidative stress by promoting the release of more harmful pro-oxidants in the target tissues [51]. The interaction of these pro-oxidants and other free radicals with nucleic acid precursors such as purine bases may increase the activity of XO in the target tissues, leading to the transformation of hypoxanthine and xanthine into their end-product, urea acid, with concomitant release of O_2_^−^. While O_2_^−^ increases the hepatorenal system’s oxidative and nitrosative stress burdens, uric acid acts as a danger-associated molecular pattern (DAMP) to activate several inflammasome pathways and NF-κB within the hepatorenal system and orchestrate the expression of pro-inflammatory mediators [52,53]. As a biomarker of hepatorenal derangements, an increase in the level of NO in the tissues of rats correlates with prolonged oxidative stress and inflammation [44,54]. Under the basal condition, NO mediates an anti-inflammatory effect and regulates several biochemical processes, including vasodilation in the cardiovascular system, neurotransmission in the central nervous system, and cytokine-mediated activation of macrophages in the immune system [55]. However, NO may accumulate and induce chronic inflammation in tissues exposed to oxidative and inflammatory stimuli. In addition, loss of redox homeostasis can trigger an increase in NO synthase activities, an enzyme that converts L-arginine to L-ornithine and NO in the presence of reduced nicotinamide adenine dinucleotide phosphate (NADPH) and molecular O_2_ as co-factors. As NO accumulates in the hepatorenal system, it interacts with O_2_.- and produces another toxic free radical, peroxynitrite (ONOO^−^). These reactive nitrogen species (RNS) mediate chronic information through DNA damage, increased Cox-2 expression, elevation in the expression of pro-inflammatory cytokines, angiogenesis, and nitration of functional proteins in the hepatorenal tissues [21,56]. In our findings, AFB_1_ orchestrated inflammation by increasing the hepatic and renal activities of MPO and XO and the levels of NO and IL-1β while decreasing the hepatic and renal levels of IL-10, an anti-inflammatory cytokine. These changes alter the pro-inflammatory/anti-inflammatory ratio favouring pro-inflammatory cues, thus increasing hepatic and renal inflammation. The induction of oxidative stress and subsequent inflammation creates suitable links between oxidative stress and inflammation. Alteration in the pro-inflammatory/anti-inflammatory ratio following exposure to AFB_1_ was reverted and normalised in cohorts of rats treated with API-containing SBE-HP as hepatic and renal MPO, XO, NO, and IL-1β in these rats significantly waned while that of IL-10 increased markedly. These alterations may have been possible due to bioactive compounds’ anti-inflammatory and antioxidative effects in the SBE-HP [30,36,57].

If the pro-oxidants resulting from the exposure to AFB_1_ are not scavenged, the resultant pro-inflammatory cues in rats’ liver and kidney tissues will not resolve. Signalling networks impairing mitochondrial dysfunction and committing cells to apoptosis may be switched upstream and downstream [58,59,60]. Specifically, free radicals produced during the metabolism of AFB_1_ and activating pro-inflammatory cells may bind to DNA and functional proteins to trigger the formation of DNA adducts in rats’ hepatic and renal tissues [22], causing genomic instability and committing the hepatocytes and nephrons to short-term cell cycle arrest. The duration of the cell cycle arrest will depend on when the AFB_1_ insults are completely detoxified and removed. However, programmed cell death may ensue if these insults persist in the liver and kidney cells. Specifically, p53 directs the expression of PUMA (p53 upregulated modulator of apoptosis), which alters the Bax/Bcl-2 ratio in favour of Bax [61]. The translocation of Bax into the hepatic and renal cells mitochondria causes cytochrome C release into the cytoplasm, where it interacts with apoptotic peptidase activating factor 1 (APAF-1) pro-caspase-9, forming apoptosome and active caspase-9, the initiator of apoptosis. Caspase-9 then activates caspase-3, the executioner of apoptosis, thereby committing the cells to die. Based on this syllogism, previous studies infer that increased activities of Caspase-9 and -3 in hepatic and renal tissue are an indication of programmed cell death of the affected tissues [22,23,62,63], and this was validated in the current studies where we observed that ingestion of AFB1 to rats significantly increased the activities of caspase-9 and -3. Any compound that can dampen the actions of these caspases could possess anti-apoptotic activity and might be used to construct a novel biologic against AFB intoxication.

Interestingly, the current study’s findings reveal that co-treatment of rats exposed to AFB_1_ with API-containing SBE-HP significantly decreased the activities of caspase-9 and -3, indicating an interruption in AFB1-mediated apoptosis and restoration of wholeness to the hepatorenal system. The role of bioactive compounds in *S. bicolor* on apoptosis depends on cell types. In normal hepatic and renal tissues, *S. bicolor* mediates the anti-apoptotic effect, thereby preserving the tissues from environmental toxicants capable of committing a cell to apoptosis. However, in tumours, these plant extractives portend a pro-apoptotic function, thus enhancing the death of cancer cells [30,39,64].

Alterations in the typical histological features of the liver and kidney tissues are clinically relevant in assessing the toxicities of AFB_1_. It also validates the outcomes of numerous biochemical assays, as established in this study. During oxidative stress, inflammation, and apoptosis, DAMPs are released into the cells, interacting with pattern recognition receptors expressed on the surface of pro-inflammatory cells [65,66]. The Kupffer cells and intraglomerular mesangial cells activate, proliferate, and differentiate into pro-inflammatory phenotypes in the liver and kidney. These cells release abundant soluble factors, including IL-1β, TNF-α, IL-17, IL-6, IL-8, NO, Cox-2, and PGE_2,_ into the hepatocytes and nephrons, thus making the cell lose their normal histology [30], as seen in the current study. However, this effect was minimised by SBE-HP through the resolution of inflammation, inhibition of oxidative stress, and apoptosis, further validating the plant’s antioxidative, anti-inflammatory, and anti-apoptotic effects.

## 4. Materials and Methods

### 4.1. Chemicals, Reagents, and Kits

AFB_1_, thiobarbituric acid (TBA), 2’, 7′-dichlorodihydrofluorescin diacetate (DCFH-DA), 5’, 5’-dithiobis-2-nitrobenzoic acid (DTNB), 1-chloro-2,4-dinitrobenzene (CDNB), hydrogen peroxide (H_2_O_2_), potassium chloride (KCl), trichloroacetic acid (TCA), sodium azide, glutathione (GSH), epinephrine, sulphosalicylic acid, xanthine, Griess reagent, and O-dianisidine were bought from Sigma-Aldrich Chemical (St. Louis, MO, USA). Biomarkers of hepatic and renal functions such as alanine aminotransferase (ALT), aspartate aminotransferase (AST), alkaline phosphatase (ALP), urea, and creatinine were purchased from Randox^TM^ Laboratories Limited, (Crumlin, UK). Enzyme-linked immunosorbent assay (ELISA) kits for interleukin 1-beta (IL-1β), interleukin-10 (IL-10), caspase-9, and caspase-3 were purchased from Elabscience Biotechnology Company (Wuhan, China).

### 4.2. Collection, Identification, and Processing of Plant Sample

Overall, 5 kg dried *S. bicolor* sheaths were purchased from Bodija Market, Ibadan, Oyo State, Nigeria. The samples were brought to the Department of Botany, the University of Ibadan, Ibadan, Nigeria, in a polythene bag for taxonomist identification, and a voucher specimen (Accession No: UIH-23118) was deposited there for reference purposes. The plant was sorted to remove dirt and other extraneous materials and ground into fine powder.

### 4.3. Extraction and Phytochemical Characterisation of S. bicolor

SBE-HP was obtained as a CH_2_Cl_2_ extract of *S. bicolor*, as described previously [30]. The isolated brownish-red paste was analysed by thin-layer chromatography (TLC) (Supelco, TLC Silica gel 60 F_254_, 1.05554.0001; Sigma-Aldrich) and LC-MS (Figure 2A,B). For LC-MS analysis, 0.6 mL isopropanol, one scoop of 0.5 mm glass beads, and one scoop of 2 mm ZrO beads were added to the SBE-HP sample in a microcentrifuge vial. The mixture was homogenised in a Tissuelyzer II at 30 Hz for 5 min, vortexed, centrifuged at 21,100× *g* for 5 min, and transferred to an LC vial for analysis. Chromatography was performed on a Waters BEH C18 100 × 2.1 mm, a 1.7-micron column with Mobile Phase A: 60%acetonitrile 40%water 0.1% FA 10 mM ammonium formate and Mobile Phase B: 10% ACN 90% IPA, 0.1% FA 10 mM ammonium formate. The column temperature was maintained at 60 °C, and a 0.5 µL sample was injected for each run. MS data were acquired in positive and negative ion modes.

### 4.4. Animal Welfare, Sample Size Estimation, and Experimental Design

The 3R’s protocols, including replacement, reduction, and refinement, were adopted in this study. The sample size was estimated using the G*Power software version 3.1.9.4 [67]. With an effect size of 0.40, 95% power, and 0.05 *p*-value for one-way analysis of variance (ANOVA) [42], a sample size of 125 was estimated. Of this calculated value, 24 adult male Wistar Albino rats (consisting of *n* = six rats, i = 4) with a mean weight of approximately 165 g b.w. were bought from the experimental animal facility of the Faculty of Veterinary Medicine, University of Ibadan, Nigeria. The experimental rats were kept in the animal facility of the Department of Biochemistry, Faculty of Basic Medical Sciences, University of Ibadan, Nigeria, and humanely maintained in natural photoperiod conditions. Rats were fed with standard chows, allowed access to clean water ad libitum, and then acclimatised for 14 days before dosing with AFB_1_ and the hydrophobic *S. bicolor* extract (SBE-HP). The study was conducted in line with the ethics of animal use as certified by the Animal Care and Use Research Ethics Committee (ACUREC) of the University of Ibadan (Approval number: UI-ACUREC/032-0521/7).

The different cohorts of rats were exposed to 28 days of repeated treatment with AFB_1_ and SBE-HP (Figure 3A). AFB_1_ and the SBE-HP stocks for dosing were prepared each day and used for intubation *per os* (*p.o*). The AFB_1_ (50 µg/kg) dose used in this study was determined from previous reports [24,68], while the SBE-HP amount was extrapolated from a dose-response study [30]. Experimental rats were grouped as designated below:

**Control:** 0.05% Carboxymethyl cellulose (CMC);

**AFB_1_ alone:** Aflatoxin B_1_, 50 µg/kg;

**AFB_1_+SBE-HP-D_1_:** AFB_1_ (50 µg/kg) + SBE-HP-D_1_ (5 mg/kg);

**AFB_1_+SBE-HP-D_2_:** AFB_1_ (50 µg/kg) + SBE-HP-D_2_ (10 mg/kg).

CMC (0.05%) was used as a vehicle for the extracts. SBE-HP was prepared by dissolving in 0.05% CMC [69]. Control animals received 0.05% CMC (0.32 mL per rat), animals in AFB_1_ alone group received 0.16 mL AFB_1,_ while animals in AFB_1_+SBE-HP-D_1_ and AFB_1_+SBE-HP-D_1_ received 0.16 mL and 0.32 mL of SBE-HP, respectively. Treatments were carried out between 09:30 and 11:30 h on the designated dates for 28 days. The 28-day treatment protocol was based on earlier experimental design and findings from our previous studies [22,30].

### 4.5. Termination of the Experiment, Organ Harvest, and Tissue Processing

At the expiration of 28 d, all the rats’ final body weight was measured 24 h after the last intubation, followed by exsanguination through the retro-orbital venous plexus system into well-labelled non-heparinised tubes. Afterwards, the rats were sacrificed through cervical dislocation. The whole blood was allowed to clot at 25 °C for 30 min, and the coagulated blood was subjected to centrifugation at 3000× *g* for 10 min at 4 °C. The clear supernatant (serum) was transferred into clean labelled Eppendorf tubes and preserved at −20 °C before quantifying the biomarkers of hepatorenal functions. In addition, the liver and kidney of the sacrificed rats were instantly removed, cleaned in ice-cold KCl solution, and then weighed using a USS-DBS16 Analytical Balance (Cleveland, OH, USA). The relative organ weights of the liver and kidney were determined as follows:$$\mathrm{Organ}\;\mathrm{weight}\;\mathrm{Relative}\;\mathrm{to}\;\mathrm{the}\;\mathrm{bodyweight}=\frac{\;\mathrm{Weight}\;\mathrm{of}\;\mathrm{organ}\;\left(g\right)}{\mathrm{Weight}\;\mathrm{of}\;\mathrm{the}\;\mathrm{body}\;\left(g\right)}\times 100.$$

Portions of the liver and kidney were used for biochemical and histological investigations. Samples for biochemical estimations were prepared by homogenising in a phosphate buffer (0.1 M, pH 7.4). The tissue homogenate was prepared by homogenising the liver (2 g in 8 mL of phosphate buffer) and the kidney (1.12 g in 4 mL of phosphate buffer) using a glass-Teflon homogeniser. The resultant homogenates were centrifuged at 12,000 rpm at 4 °C for 15 min to obtain a clear mitochondrial fraction. The supernatants were collected in aliquots and frozen before quantifying oxidative, inflammatory, and apoptotic biomarkers.

### 4.6. Estimation of Function and Integrity of the Hepatorenal System

The functionality and integrity of the liver and kidney of rats co-exposed to AFB_1_ and SBE-HP after 28 d was evaluated by measuring the activities and levels of ALT, AST, ALP, creatinine, and urea using commercial kits as previously reported [22].

### 4.7. Estimation of the Biomarkers of Oxidative Stress, Inflammation, and Apoptosis

The oxidative stress, inflammation, and apoptosis levels were measured in the liver and kidney of rats co-exposed to AFB_1_ and SBE-HP post 28 d. The levels of total protein in the liver and kidney of rats were estimated according to the method described by ref. [46]; hepatic and renal activities of superoxide dismutase (SOD) were assessed by the methods described by Misra and Fridovich [70], as previously reported [71]; hepatic and renal activities of catalase (CAT) were assessed by the protocols of Clairborne [72] using H_2_O_2_ as a substrate as previously reported [73]; hepatic and renal activities of Glutathione-S-transferase (GST) and glutathione peroxidase (GPx) were quantified by the procedures of Habig [74] and Rotruck et al. [75], respectively, as previously reported [76]; hepatic and renal levels of reduced glutathione (GSH) and total sulfhydryl group (TSH) were estimated by the method of Jollow et al. [77] and Ellman [78], as previously reported [22,57]; hepatic and renal activities of xanthine oxidase (XO) were measured by the procedures of Bergmeyer et al. [79]; hepatic and renal levels of malondialdehyde (MDA), otherwise termed lipid peroxidation (LPO) were assessed by the method described by Okhawa [80]; hepatic and renal levels of reactive oxygen and nitrogen species (RONS) were assayed by the protocols of Owumi and Dim [81]; hepatic and renal nitric oxide (NO) level and myeloperoxidase (MPO) activity were quantified by the protocols of Green et al. [60,82] and Granell et al. [83,84], respectively, as previously reported by Owumi et al. [85]; hepatic and renal levels of IL-1β and IL-10 and the caspase-9 and -3 activities were assayed using ELISA kits, as previously reported [22]. All measurements were carried out using a Spectra Max^TM^ plate reader.

### 4.8. Examination of the Histological Sections of the Liver and Kidney

The portions of the liver and kidney for histological assessment were fixed in neutral buffered formalin (10%) before the preparation of histological sectioning and staining. With the aid of a standard paraffin-wax method, the hepatic and renal tissues were processed for histopathological examination in line with the description of Bancroft and Gamble [86]. Approximately five μm thickness of the portion of the liver and kidney were dyed with haematoxylin and eosin and processed for light microscopy. All prepared slides were coded and probed with a Carl Zeiss Axio light microscope (Gottingen, Germany). On inspection, images were taken using a Zeiss Axiocam 512 camera (Gottingen, Germany) attached to the microscope by a pathologist unaware of the various treatment cohorts from which the slides were prepared.

### 4.9. Statistical Analysis of Results

At the end of the experiments, data were generated, quantified, and subjected to statistical analyses using quantitative measures such as mean and standard deviation. The results were expressed as the mean ± SD of replicates. A test of statistical inferences was performed by student t-test to compare the significance between the initial body weight (IBW) and final body weight (FBW) of rats. A one-way analysis of variance (ANOVA) followed by a post hoc test (Tukey’s test) set at a 95% probability level was used to test the significance difference across the four experimental groups using GraphPad Prism, version 8.3.0 for Mac (www.graphpad.com, accessed on 9 July 2022; GraphPad, San Diego, CA, USA).

## 5. Conclusions

We have shown that API-containing SBE-HP prevented, in adult male Wistar Albino rats, oxidative/nitrosative stress, inflammation, and apoptosis induction. The probable mechanisms may be activating the aryl hydrocarbon receptor (which promotes IL-10 expression), de-repression of estrogen receptor alpha (known to mediate anti-inflammatory activity), activation of the expression of p53 and mitochondrial membrane potential, and upregulation of Nrf2, NQO1, and HO-1 signalling. At the same time, mediating the suppression of cytochrome P450 (CYP) 1A2 expression, inhibition of NF-kB, upregulation of Bax expression and the activities of caspase-3 and caspase-3 and suppression of signalling along JNK, p38 MAKP, ERK, and Keap1 axis as depicted in our proposed mechanism of protection Figure 12. These findings recapitulate the ethnomedicinal relevance of the SBE-HP and emphasise its usefulness in preventing AFB_1_ intoxications, including hepatic steatosis, hepatitis, cirrhosis, hepatocellular carcinoma, and glomerulonephritis in animal and human models. Finally, our studies validate the three defined mechanisms of AFB_1_ intoxications, including oxidative stress, inflammation, and apoptosis, and warrant further investigation to explore more mechanistic pathways of AFB1 intoxication in rats and other experimental models.

## Figures and Tables

**Figure 1 molecules-28-03013-f001:**
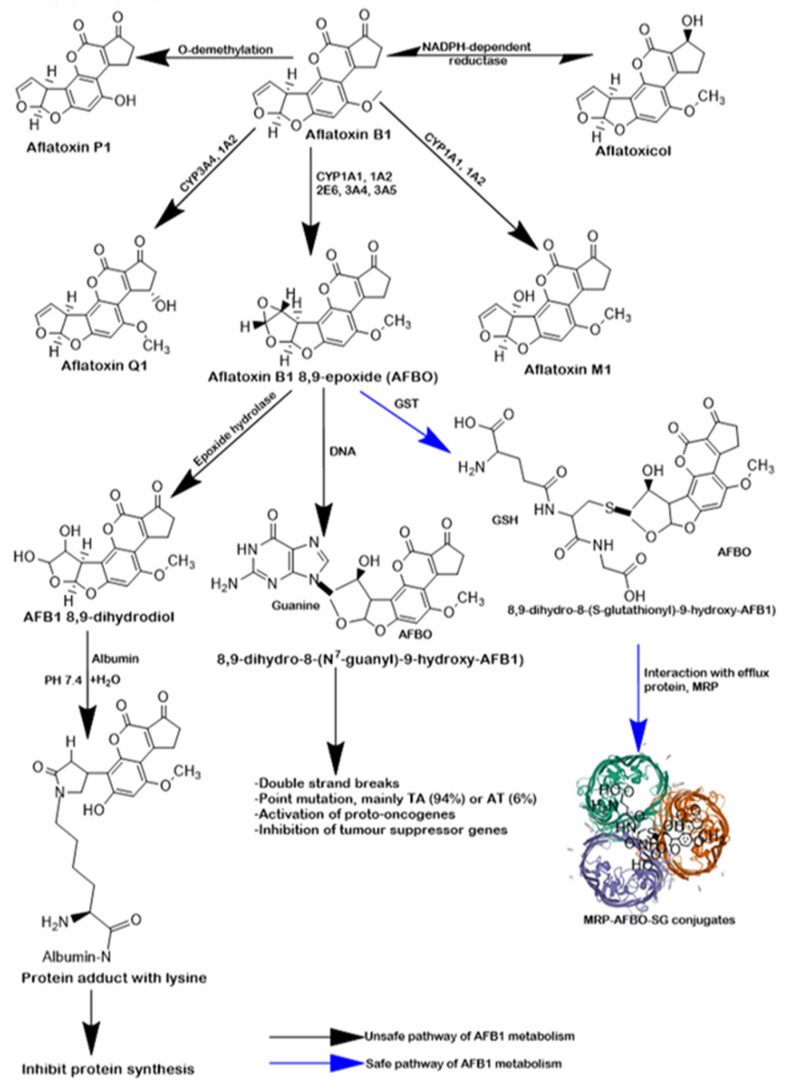
Metabolism of AFB1. Following chronic exposure to AFB1, AFB1 is rapidly absorbed into the enterocytes by passive diffusion and is absorbed into the liver for metabolism by arrays of CYP isoforms. Specifically, NADPH-dependent reductase converts AFB1 into aflatoxicol (AFL), CYP3A4 and CYP1A2 convert AFB1 into aflatoxin Q1 (AFQ1), CYP1A1 and CYP1A2 convert AFB1 into aflatoxin M1 (AFM1), or aflatoxin P1 (AFP1) through an O-demethylation reaction. These intermediates are clinically irrelevant as they are not heavily implicated in mutagenicity, carcinogenicity, and teratogenicity. The hepatic expression of CYP1A1, CYP1A2, CYP2E6, CPY3A4, and CYP3A5 is associated with AFBO formation, a clinically relevant intermediate, implicated in AFB_1_ toxicity and carcinogenicity. The mechanisms of AFBO toxicity are through the formation of protein adducts, DNA adducts, and lipid peroxidation. Specifically, epoxide hydrolase at PH 7.4 hydrolyses AFBO to AFB_1_ 8,9-dihydrodiol, which can form a Schiff’s base with primary amino groups in lysine residues, forming a protein adduct with lysine. This adduct is known to inhibit protein synthesis. In addition, AFBO binds to the guanine of DNA and forms a DNA adduct known as 8,9-dihydro-8-(N^7^-guanyl)-9-hydroxy-AFB1, leading to double-strand break, point mutation, activation of pro-oncogenes, and suppression of tumour suppressor genes. There is a safe pathway (blue arrow) for degrading AFBO, mediated by GST, an enzyme that mediates the conjugation of AFBO and GSH to form the AFB1-*S*-G complex. MRP binds to this complex and removes them from the hepatocytes for excretion in the bile and urine. *Created by ChemDraw*.

**Figure 2 molecules-28-03013-f002:**
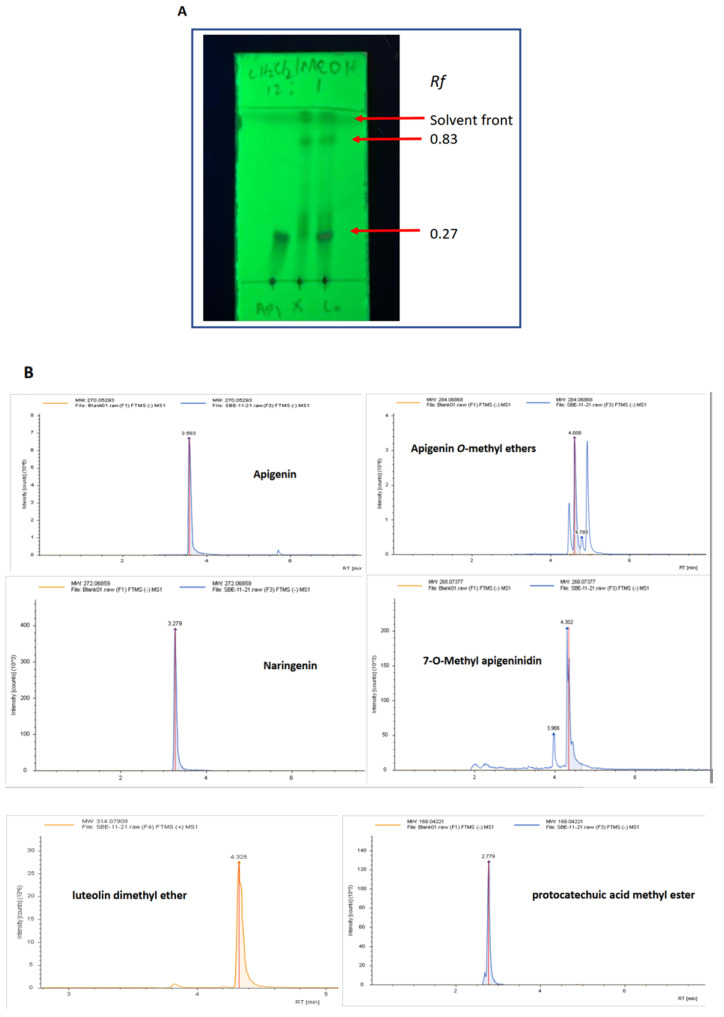
Characterisation of the key components in SBE-HP using TLC and LC-MS analyses. (**A**) TLC analysis on a normal phase silica plate, eluted with CH_2_Cl_2_/MeOH 12:1, revealed that SBE-HP separated into three clusters with *RF*s of 0.27, 0.83, and approx. 1. (**B**) LC-MS traces of key compounds in SBE-HP. Negative ion mode was presented for clarity.

**Figure 3 molecules-28-03013-f003:**
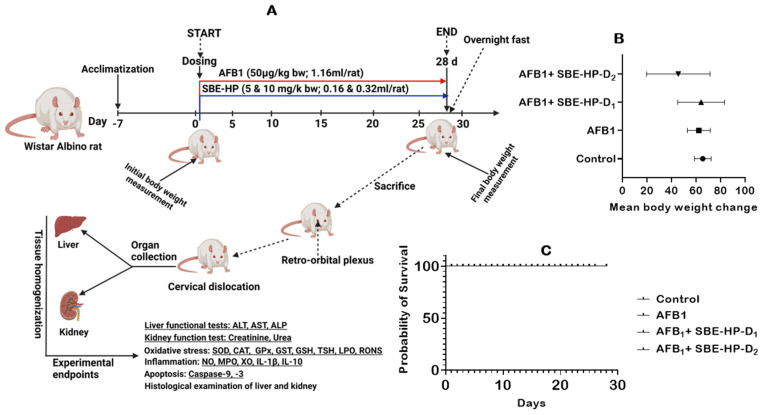
Experimental protocol of AFB_1_ and API-rich hydrophobic fraction of *S. bicolor* extracts 28 consecutive days. (**A**) In vivo screening of the hepatoprotective effect of SBE-HP on AFB1-challenged adult male Wistar Albino rats for 28 consecutive days, (**B**) effect of SBE-HP on the mean body weight change of AFB1-treated treated, and (**C**) Kaplan-Meier curve of rats treated with SBE-HP and AFB_1_. Created by https://app.biorender.com/ (accessed on 8 July 2022).

**Figure 4 molecules-28-03013-f004:**
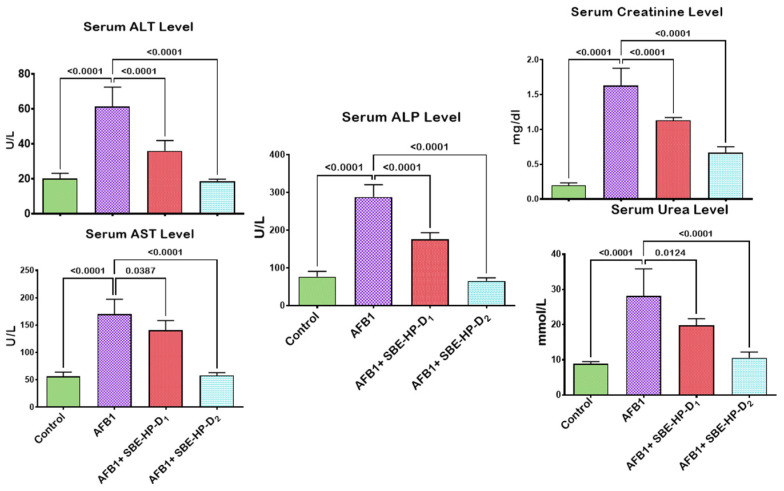
Effect of SBE-HP on the liver and kidney function of rats treated with AFB_1_ for 28 d. Experimental doses: AFB_1_ at 50 μg/kg; SBE-HP-D1 at 5 mg/kg; SBE-HP-D2 at 10 mg/kg. Values are expressed as mean ± SD for six rats per treatment cohort. Connecting lines indicate groups compared to one another. The significance level was set at (*p* < 0.05); *p* < 0.05 suggests the level of significance; *p* > 0.05: not significant. AFB_1_: Aflatoxin B1; D1: lower dose; D2: higher quantity; ALT: Alanine aminotransferase; AST: aspartate aminotransferase, ALP: Alkaline phosphatase; GGT: gamma-glutamyl transferase.

**Figure 5 molecules-28-03013-f005:**
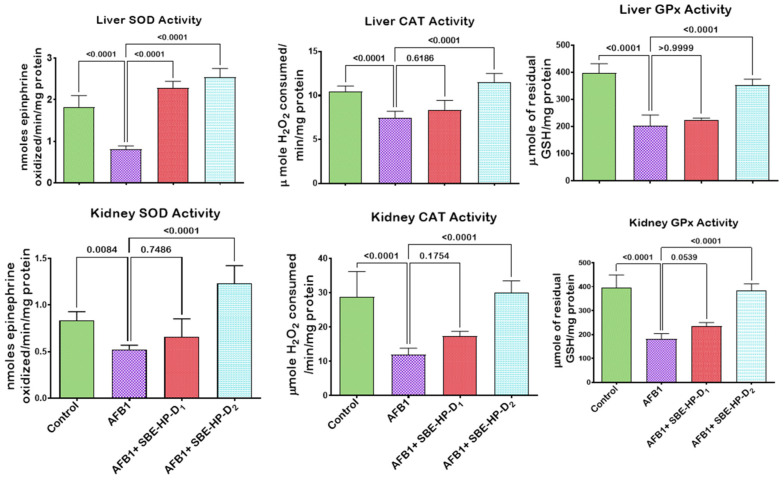
Effect of SBE-HP on the tissue concentrations of CAT, SOD, and GPx in the liver and kidney of rats treated with AFB_1_ for 28 d. AFB_1_ at 50μg/kg; SBE-HP-D1 at 5 mg/kg; SBE-HP-D2 at 10 mg/kg. Values are expressed as mean ± SD for six rats per treatment cohort. Connecting lines indicate groups compared to one another. The significance level was set at (*p* < 0.05); *p* < 0.05 indicates the level of significance; *p* > 0.05: not significant. AFB_1_: Aflatoxin B1; D1: lower dose; D2: higher dose; SOD: Superoxide dismutase; CAT: Catalase; GPx: Glutathione peroxidase.

**Figure 6 molecules-28-03013-f006:**
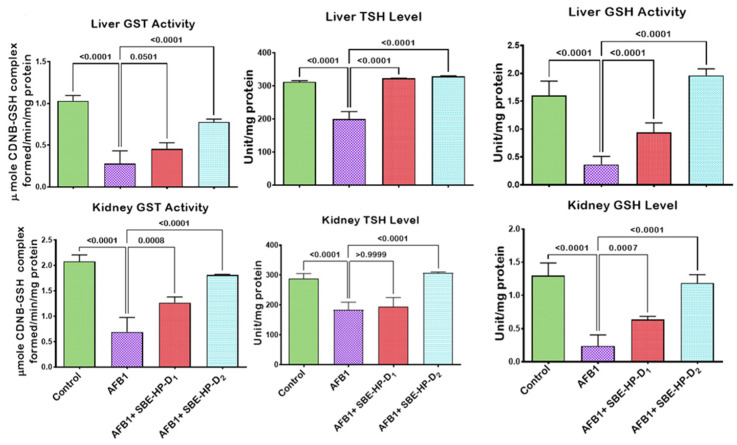
Effect of SBE-HP on the tissue concentrations of GST, GSH, and TSH in the liver and kidney of rats treated with AFB_1_ for 28 d. AFB_1_ at 50 μg/kg; SBE-HP-D1 at 5 mg/kg; SBE-HP-D2 at 10 mg/kg. Values are expressed as mean ± SD for six rats per treatment cohort. Connecting lines indicate groups compared to one another. The significance level was set at (*p* < 0.05); *p* < 0.05 suggests the level of significance; *p* > 0.05: not significant. AFB_1_: Aflatoxin B1; D1: lower dose; D2: higher dose; GST: Glutathione S-transferase; GSH: reduced glutathione; TSH: Total sulfhydryl group.

**Figure 7 molecules-28-03013-f007:**
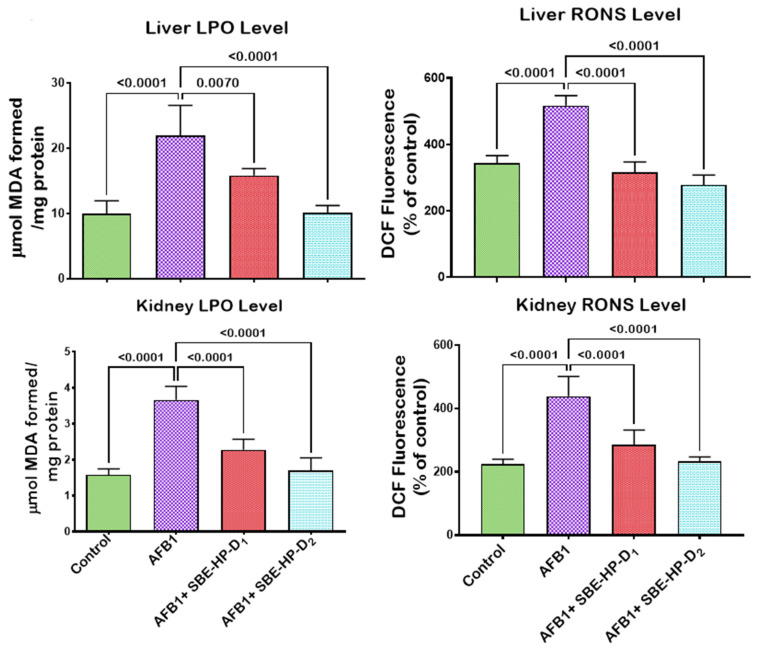
Effect of SBE-HP on the concentrations of LPO and RONS in the liver and kidney of rats treated with AFB_1_ for 28 d. AFB_1_ at 50 μg/kg; SBE-HP-D1 at 5 mg/kg; SBE-HP-D2 at 10 mg/kg. Values are expressed as mean ± SD for six rats per treatment cohort. Connecting lines indicate groups compared to one another. The significance level was set at (*p* < 0.05); *p* < 0.05 suggests the level of significance; *p* > 0.05: not significant. AFB_1_: Aflatoxin B1; D1: lower dose; D2: higher dose; LPO: Lipid peroxidation; RONS: Reactive oxygen and nitrogen species.

**Figure 8 molecules-28-03013-f008:**
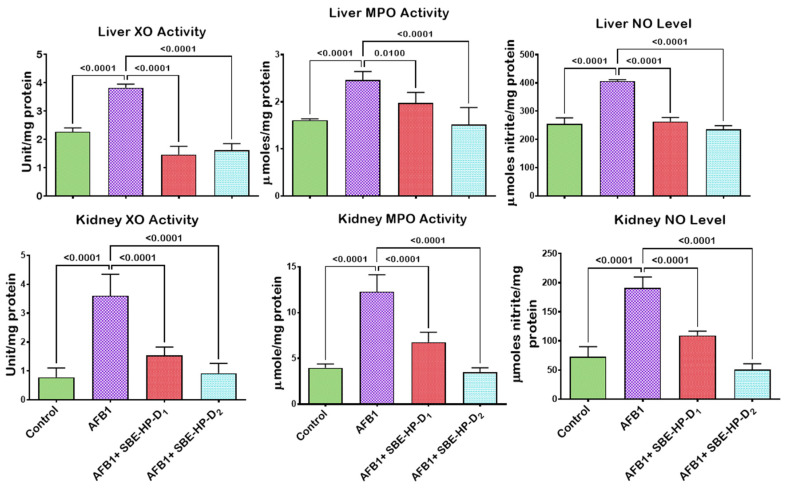
Effect of SBE-HP on the tissue concentrations of XO, MPO, and NO in the liver and kidney of rats treated with AFB_1_ for 28 d. AFB_1_ at 50 μg/kg; SBE-HP-D1 at 5 mg/kg; SBE-HP-D2 at 10 mg/kg. Values are expressed as mean ± SD for six rats per treatment cohort. Connecting lines indicate groups compared to one another. The significance level was set at (*p* < 0.05); *p* < 0.05 indicates the level of significance; *p* > 0.05: not significant. AFB_1_: Aflatoxin B1; D1: lower dose; D2: higher dose; XO: Xanthine oxidase; MPO: Myeloperoxidase; NO: Nitric oxide.

**Figure 9 molecules-28-03013-f009:**
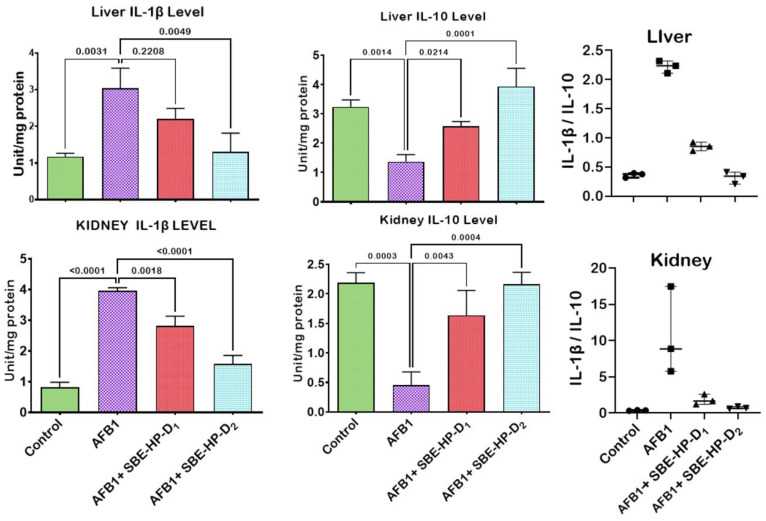
Effect of SBE-HP on tissue concentrations of IL-1β and IL-10 in the liver and kidney of rats treated with AFB_1_ for 28 d. AFB_1_ at 50 μg/kg; SBE-HP-D1 at 5 mg/kg; SBE-HP-D2 at 10 mg/kg. Values are expressed as mean ± SD for six rats per treatment cohort. Connecting lines indicate groups compared to one another. The significance level was set at (*p* < 0.05); *p* < 0.05 suggests the level of significance; *p* > 0.05: not significant. AFB_1_: Aflatoxin B1; D1: lower dose; D2: higher dose; IL-1β: Interleukin-1beta; IL-10: Interleukin-10.

**Figure 10 molecules-28-03013-f010:**
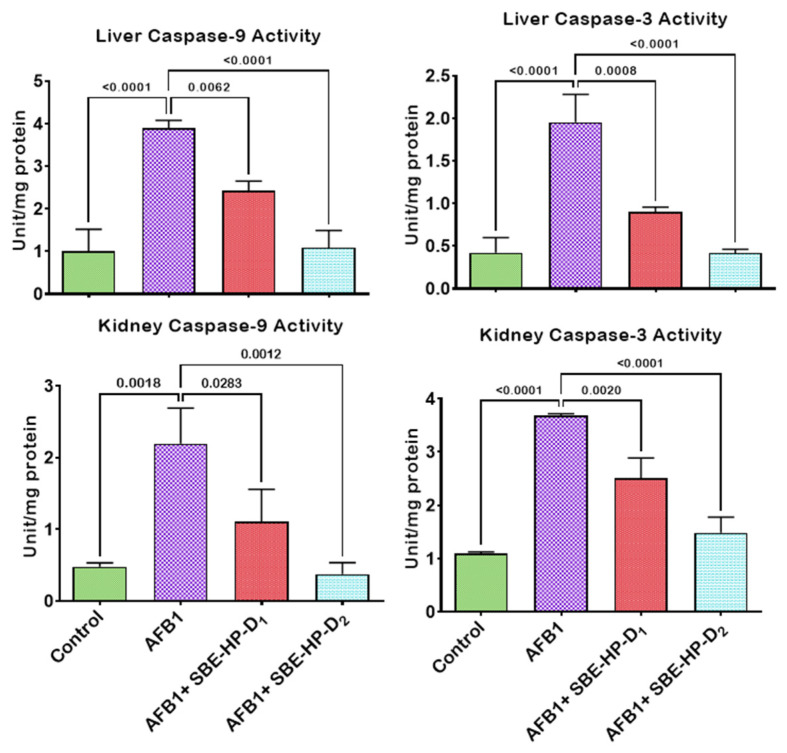
Effect of SBE-HP on tissue concentrations of caspase-9 and caspase-3 in the liver and kidney of rats treated with AFB_1_ for 28 d. AFB_1_ at 50 μg/kg; SBE-HP-D1 at 5 mg/kg; SBE-HP-D2 at 10 mg/kg. Values are expressed as mean ± SD for six rats per treatment cohort. Connecting lines indicate groups compared to one another. The significance level was set at (*p* < 0.05); *p* < 0.05 suggests the level of significance; *p* > 0.05: not significant. AFB_1_: Aflatoxin B1; D1: lower dose; D2: higher dose.

**Figure 11 molecules-28-03013-f011:**
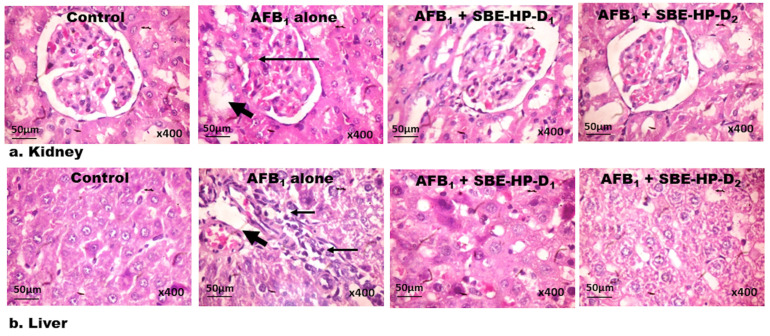
(**a**) Control Plates of the kidney show a focal area of mild congestion and apoptotic bodies with typical tissue architecture. AFB_1_ alone plates show disseminated glomerular messangialisation (thin arrow) and the extent of infiltration of the cortex by inflammatory cells (bold arrowhead). SBE-HP alone plate (not shown) tissues appear normal and relatively like those from control tissue sections. AFB_1_ with SBE-HP-D1 and SBE-HP-D2 plates dose-dependently improved histo-architecture of the kidney with the mild presence of inflammatory cells. (**b**) AFB_1_ alone shows areas of focal congestion (bold arrows), infiltration of zone 2 by inflammatory cells, mild hydropic/ballooning degeneration of the hepatocytes, and moderate microvesicular steatosis (tiny arrows). SBE-HP alone (plate not shown) tissue morphologies are similar to the control plate. AFB_1_ with SBE-HP-D1 and SBE-HP-D2 plates improved hepatic cytoarchitecture with mild focal congestion and infiltration of zone 2 by inflammatory cells. H and E-stained sections; magnification at ×400.

**Figure 12 molecules-28-03013-f012:**
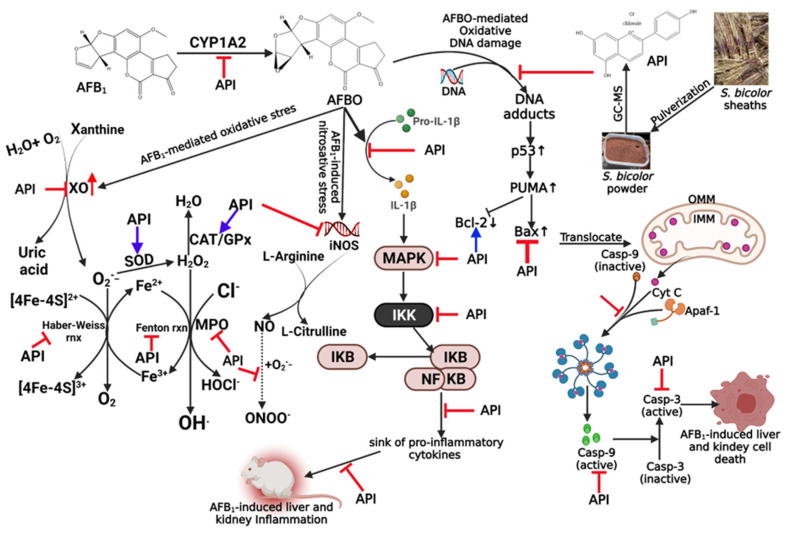
Proposed mechanism of SBE-HP ameliorative effect on AFB_1_-mediated toxicities in the liver and kidney of an experimental rat model. SBE-HP prevented AFB_1_-induced oxidative and nitrosative stress and inflammation by reducing the activity of CYP1A2, NF-kB-mediated generation of pro-inflammatory cytokines, and IL-1β. SBE-HP also reduced apoptosis by altering the Bcl-2/Bax ratio favouring caspase 9 and caspase 3 activity. Created by https://app.biorender.com/ (accessed on 9 July 2022).

## Data Availability

The datasets used and analysed during the current study are available from the corresponding author upon reasonable request.

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
