# Peer review of "The Hydrophobic Extract of Sorghum bicolor (L. Moench) Enriched in Apigenin-Protected Rats against Aflatoxin B1-Associated Hepatorenal Derangement"

_molecules, 2023, doi:10.3390/molecules28073013_

Round 1
Reviewer 1 Report
It is an interesting study demonstrating that AFB1-induced oxidative stress, inflammation, and apoptosis in the hepatorenal system of rats could be prevented by treating rats with Sorghum bicolor sheath hydrophobic extract (SBE-HP).
Comments:
- Why did the authors choose the 50 μg/kg aflatoxin concentration for their study? In the Material and Methods section, the authors stated: “The AFB1 (50 μg/kg) dose used in this study was determined from previous reports [41, 42].” However, in the study [41], the 75 μg/kg dose was used for experiments. Unexpectedly, the study [42] does not even mention aflatoxin. Please explain.
- It appears that the effects of SBE-HP are dose-dependent. However, this observation should be checked by statistical analysis to determine the significance of differences between AFB1+SBE-HP-D1 and AFB1+SBE-HP-D2 treatments. Results should be presented and discussed.
- The authors should statistically compare the data obtained in AFB1+SBE-HP-D1 and SBE-HP-D2 treatments with those obtained in untreated control to evaluate whether the doses used are sufficient to restore the investigated parameters to levels similar to those observed in control. Results should be presented and discussed.
Author Response
The Authors have provided a point-by-point response to the Reviewer's comment as a PDF attachment below. Thank you

Reviewer 2 Report
There is no doubt that this is a sound paper. The data in this paper are sufficient, the logic is rigorous, and the results have potential application value. I recommend accepting this manuscript for publication as it is.
Author Response

(The authors gave the same response as above.)
